# Partnership Status and Living Situation in Persons Experiencing Physical Disability in 22 Countries: Are There Patterns According to Individual and Country-Level Characteristics?

**DOI:** 10.3390/ijerph17197002

**Published:** 2020-09-24

**Authors:** Christine Fekete, Mohit Arora, Jan D. Reinhardt, Mirja Gross-Hemmi, Athanasios Kyriakides, Marc Le Fort, Julia Patrick Engkasan, Hannah Tough

**Affiliations:** 1Swiss Paraplegic Research, 6207 Nottwil, Switzerland; reinhardt@scu.edu.cn (J.D.R.); mirja.gross@paraplegie.ch (M.G.-H.); hannah.tough@paraplegie.ch (H.T.); 2Department of Health Sciences & Medicine, University of Lucerne, 6002 Lucerne, Switzerland; 3John Walsh Centre for Rehabilitation Research, Kolling Institute of Medical Research, Royal North Shore Hospital, St Leonards, NSW 2065, Australia; mohit.arora@sydney.edu.au; 4Sydney Medical School-Northern, Faculty of Medicine and Health, The University of Sydney, Sydney, NSW 2006, Australia; 5Institute for Disaster Management and Reconstruction of Sichuan University and Hong Kong Polytechnic University, Sichuan University, Chengdu 610207, China; 6Spinal Cord Injuries Unit, University of Patras, Rio, 26500 Patras, Greece; athkyriaky@yahoo.com; 7Neurological Physical and Rehabilitation Medicine Department, University Hospital Saint-Jacques, 44093 Nantes, France; marc.lefort@chu-nantes.fr; 8Department of Rehabilitation Medicine, University of Malaya, Kuala Lumpur 50603, Malaysia; julia@ummc.edu.my

**Keywords:** marital status, partnership, living alone, household composition, disability, spinal cord injury

## Abstract

Persons experiencing disabilities often face difficulties to establish and maintain intimate partnerships and the decision whether to live alone or with others is often not their own to make. This study investigates whether individual and country-level characteristics predict the partnership status and the living situation of persons with spinal cord injury (SCI) from 22 countries. We used data from 12,591 participants of the International SCI Community Survey (InSCI) and regressed partnership status and living situation on individual (sociodemographic and injury characteristics) and country-level characteristics (Human Development Index, HDI) using multilevel models. Females, younger persons, those with lower income, without paid work, more severe injuries, and longer time since injury were more often single. Males, older persons, those with higher income, paid work, less severe injuries, and those from countries with higher HDI more often lived alone. This study provides initial evidence for the claim that the partnership status and the living situation of people with SCI are influenced by sociodemographic and socioeconomic factors and are not merely a matter of choice, in particular for those with severe injuries.

## 1. Introduction 

Persons experiencing disabilities often face constraints in establishing and maintaining intimate partnerships due to attitudinal barriers and restricted opportunities to meet potential partners [1]. Not only are persons living with disabilities confronted with negative societal attitudes and discrimination that limit their opportunities to enter and form lasting partnerships, but they may also hold internalized stigmatizing beliefs, such as feelings of inferiority and beliefs of not deserving intimacy, sexuality, and love [2,3,4,5]. Persons living with disabilities are less likely to be married or have an intimate partnership than persons without a disabling condition [1,6,7,8,9,10], and if they marry, they are at increased risk of separation and divorce [1,11,12,13,14]. People with an early onset of a disabling condition, cognitive impairment, mobility limitations, unemployment, and lower levels of education seem particularly disadvantaged in finding a partner [1,6].

Besides restricted opportunities in forming partnerships, disability may also affect people’s living situation, since the choice where and with whom to live is often not self-determined but limited by socioeconomic and health-related resources or the lack of stability in a relationship [8]. Furthermore, restrictions in people’s freedom of choice can lead to both being unable to live independently in a one-person household or being excluded from living with others. For example, not finding roommates that wish to share an apartment or the lack of a partnership or parenthood [1,6,7,8,15] may lead to the involuntary outcome of living alone. A recent study from Sweden documented that individuals experiencing disabilities are as twice as likely to live alone and to report longer periods of living alone than people without disabilities [9]. Conversely, difficulties in living alone in an independent household may be related to limited functional capacity, unavailability of informal support, and lack of adapted housing or assistive devices. This may lead to the situation that people have to live with family caregivers or in institutions such as sheltered accommodation. Investigations into the living situation of people with disabling conditions need to take into account broader societal conditions, as customs related to the living situation vary regionally; for example, living alone is generally less common in Asian as compared to European countries [16]. Moreover, the availability of formal and informal care outside the family might be limited in some settings leading to the necessity for persons experiencing disabilities to live with family who can provide informal care [17,18]. 

In the present study, we examine the above outlined issues for people with spinal cord injury (SCI). SCI is a physical impairment resulting from a traumatic or non-traumatic injury to the spinal cord that affects motor, sensory, and autonomic functions at and below the neurological level of injury. SCI often markedly impacts on people’s health and functioning in daily life [19] and can be viewed as a model for physical disability of different kinds. Previous studies in persons with SCI have mainly focused on factors that influence separation or divorce [12,13,14] and not on the more general question about factors predicting marital or partnership status. Results from empirical research on the living situation of people with SCI are currently unavailable. Apart from contributing to closing these gaps, our study can draw on data from persons with SCI that have been collected in 22 countries using the same questionnaire. It thus provides the unique opportunity to study the influence of country-level characteristics on partnerships and living situation in addition to individual factors. Specifically, this study aimed to investigate associations of individual (sociodemographic and SCI characteristics) and country-level characteristics (Human Development Index, HDI) with the partnership status and living situation of persons with SCI from different regions of the world. 

## 2. Materials and Methods

### 2.1. Design 

The cross-sectional InSCI community survey was implemented in 22 countries from all continents between 01/2017 and 05/2019 to describe aspects of functioning, health, and well-being of persons with SCI [20,21]. The networks of the International Society of Physical Rehabilitation and Medicine and the International Spinal Cord Society (ISCoS) were used to establish collaboration between countries. National representatives closely involved in SCI research or clinical practice firstly evaluated the feasibility of implementing a survey in their country and then formally joined the InSCI network at the kick-off meeting in February 2015 in Switzerland to initiate collaboration. A standardized questionnaire consisting of 125 items assessing various aspects of living with SCI was used as described in more detail elsewhere [22]. At least 18-year-old, community-dwelling people with traumatic or non-traumatic SCI, who were able to respond in an available questionnaire language version were eligible for participation. People with congenital etiology, neurodegenerative disorders (e.g., multiple sclerosis) or Guillain Barré syndrome were excluded [21], as people with these health conditions usually follow different rehabilitation paths and disease progressions than those with acquired SCI. National Study Centers were responsible for recruitment, data collection, and the organization of resources to execute the survey. Sampling frames were defined according to local conditions, including convenience or random sampling. Random sampling based on predefined sampling frames was only possible in 8 countries that had access to hospital or patient organization databases. Due to a lack of access to such databases, 14 countries used convenience sampling methods and recruited individuals visiting health care facilities or joining patient organization events [23]. As indicated by a power analysis, countries were requested to recruit a minimum of 200 participants [21]. A total of 10 countries recruited between 200 and 300 participants as detecting potential participants was challenging because SCI is a rare condition and systematic records were missing or relevant databases were not accessible. Recruitment outcomes of the different countries are reported elsewhere in more detail [23]. Countries offered paper–pencil or online questionnaires and telephone or personal interviews as local circumstances permitted. Ethical approval was obtained from Institutional Review Boards or Ethical Committees in all countries and the study was conducted according to the principles of the Helsinki Declaration (EKNZ, 11042 PB_2016-02608). Informed consent was obtained from each participant and participants could withdraw from the study at any time. Further details on recruitment and data collection procedures, response rates, and a profile of the InSCI population are provided elsewhere [21,23]. 

### 2.2. Measures

#### 2.2.1. Outcomes

Partnership status was assessed with a question asking for current marital status (single; married; cohabiting or in a partnership; separated or divorced; widowed). This item was dichotomized for further analysis so that the responses married and cohabiting or in a partnership were categorized as ‘having a partner’ and other responses as ‘being single’. The living situation was assessed with a question asking whether participants lived alone or not. The 413 persons who indicated living in an institution (e.g., nursing home, home for the elderly) were excluded from analysis on the living situation. 

#### 2.2.2. Predictors

##### Individual Characteristics

Sociodemographic characteristics considered here included gender, age at survey in years, education, household income, and employment status. Education was assessed in line with the International Standard Classification of Education (ISCED), summing up total years of formal education, including school and vocational training [24], and potential vocational retraining after SCI. Net-equivalent household income in the countries’ currency was calculated by including information on disposable household income, weighted by household composition (number of adults and children) according to criteria of the Organisation for Economic Co-operation and Development [25]. In order to create comparable categories taking into account country differences in educational and economic systems, we built country-based distributional quartiles for education and income to group persons within countries into four categories ranging from low to high education or income. These country-based quartiles were then converted into a variable for the total sample, whereby all people in the lowest, second lowest, second highest, and highest country-specific quartile were combined. Employment status was assessed with a multiple-choice question about the current employment situation: people who indicated working for wages or salary with an employer (also if currently on sick leave for more than three months) or being self-employed were coded as having paid work, and persons who did not cross either of those categories were coded as having no paid work. 

SCI characteristics included information on level of injury (paraplegia vs. tetraplegia), completeness of injury (complete vs. incomplete), etiology (traumatic vs. non-traumatic), and time since injury in years. We created a four-categorical variable for injury severity, combining the information on lesion type and completeness (paraplegia incomplete, paraplegia complete, tetraplegia incomplete, tetraplegia complete). 

##### Country-Level Characteristics 

Country-level characteristics were assessed with the HDI, which is a statistical composite index of country achievements in key dimensions of human development available for all 22 InSCI countries. The HDI takes into account the national life expectancy at birth, education (mean of years of schooling for adults aged over 25 years and expected years of schooling for children of school entering age), and income (gross national income per capita by purchase power parity in USD) [26]. Countries were classified based on HDI rank in the first year of data collection (2017), published on the webpage of the United Nations Development Program (hdr.undp.org/en/data). We created a variable indicating the rank within all InSCI countries according to the HDI, with higher ranks indicating higher HDI. Of note, the HDI score ranges from 1–21 and not from 1–22 as one could assume based on the number of InSCI countries, as Indonesia and South Africa have identical HDI scores and were thus placed on the same rank. The final ranking is as follows: 1 = Norway, 2 = Switzerland, 3 = Germany, 4 = Australia, 5 = Netherlands, 6 = United States, 7 = Japan, 8 = South Korea, 9 = Spain, 10 = France, 11 = Italy, 12 = Greece, 13 = Poland, 14 = Lithuania, 15 = Romania, 16 = Malaysia, 17 = Thailand, 18 = Brazil, 19 = China, 20 = Indonesia and South Africa, 21 = Morocco. Based on this ranking, we additionally created distribution-based quartiles for descriptive analysis. 

### 2.3. Statistical Analysis 

Analyses were conducted using STATA version 16.0 for Windows (Publisher: College Station, TX, USA). First, descriptive analysis of distributions of variables of interest were given for the overall sample. Second, univariable and multivariable models were run to predict the outcome of being single and living alone, respectively. In multivariable models, all individual (sociodemographic and SCI characteristics) and country-level characteristics (HDI rank) were included simultaneously. We used multilevel logistic regression with random intercept for country to account for unobserved heterogeneity due to clustering of data within countries [27]. We report odds ratios (OR) with 95% CI and *p*-values from likelihood ratio tests. Important to note, analysis on the living situation were restricted to the subsample of people who were single, because over 97% of people who were in a partnership also lived with others. 

Sensitivity analysis was performed in order to assess whether missing values would impact on study results. Therefore, missing values were imputed by multiple imputation (MI), assuming that missing values were not related to values of other observed variables (missing at random). We used MI by chained equations (MICE) to impute different types of variables, including categorical, ordinal, and continuous variables [28]. Differences between complete case analyses and analyses based on MI data indicated negligible differences between the two strategies (results not shown). Results shown are based on complete case analyses.

## 3. Results 

A description of the study sample can be found in Table 1. Forty one percent of the sampled population was single and 19% lived alone, with only two percent being in a partnership and living alone at the same time. Roughly, three-quarters were male, mean age was around 51 years (minimum age 18; maximum age 96), and around one-third of people had paid work. Incomplete paraplegia was the most common injury severity (35%), and complete tetraplegia was the least frequent (10%). Four out of five persons sustained a traumatic SCI, and one third of the sample had been living with SCI for less than five years. Mean time since injury was around 13 years and people reported on average 12 years of education. The number of study participants was higher in countries with higher HDI, with 44% being in the highest HDI quartile and 21% in the lowest. 

### 3.1. Partnership Status

Table 2 presents the results of the unadjusted and adjusted multilevel models for the association of individual and country-level characteristics with the partnership status. Unadjusted results show that females, younger participants, people without paid work, and those with lower income were more likely to be single. In contrast, people with higher education had higher odds of being single than those with lower education. Moreover, people with more severe injuries, traumatic etiology, and longer time since injury had increased odds of being single. The HDI-rank of countries was unrelated to partnership status in our sample. 

Results of adjusted multilevel models largely confirm results of unadjusted models, showing that females, younger people, those with lower income, and people without paid work were more likely to be single. The same was found for people with more severe injuries as compared to those with less severe injuries (i.e., incomplete paraplegia). The odds of being single also increased with time since injury. Etiology, education, and the HDI-level of the country were not related to partnership status. 

### 3.2. Living Situation

Unadjusted and adjusted result multilevel models for the association of individual and country-level characteristics with the living situation of singles are displayed in Table 3. Univariate analysis showed that older age, higher income, lower education, less severe injury, non-traumatic etiology, and longer time since injury were associated with living alone. The odds of living alone increased gradually with higher HDI rank, indicating that people from more developed countries were more likely to live in one-person households than people from countries with lower HDI rank if they were not in a partnership. 

Adjusted models show that among singles, males, older people, those with higher income, and less severe injuries more often reported living alone. Education, employment status, etiology, and time since injury were not associated with the living situation of singles in adjusted models. The odds of living alone increased, however, with greater HDI rank, indicating that singles from higher HDI countries had higher odds of living alone than singles from lower HDI countries. 

## 4. Discussion 

This study is the first to comprehensively describe the relation of sociodemographic and disability-related variables as well as country-level development with partnership status and living situation of persons experiencing physical disability from different regions of the world. We found clear patterns in the partnership status and living situation of people with SCI according to individual sociodemographic and SCI characteristics, as females, younger people, those with lower income, without paid work, more severe injury, and living a longer time with disability less often had a partner. Among singles, being male, having higher income, paid work, and less severe injury increased the likelihood of living alone. However, country-level development as operationalized with the HDI was only related to the living situation and not so to the partnership status as singles from countries with higher HDI markedly more often lived alone than singles from lower HDI countries. 

Our results on predictors of partnership status in SCI are largely in line with previous observations that also offer plausible explanations [1,6]. Gender differences in the vulnerability to discrimination might explain the fact that women were more often singles than men, as women living with disabilities are particularly prone to negative attitudes regarding their sexuality and ability to meet role demands of motherhood (double discrimination) [5,29]. Different patterns in participation on other social activities between males and females may additionally explain gender differences in the partnership status, as findings from a Swiss SCI sample demonstrated that females generally reported lower participation frequency and greater restrictions than males [30], thus limiting women’s opportunities to meet potential partners. Our results that people with higher income, paid work, and less severe injuries were more often in a partnership confirm previous findings from disability studies [1,6]. Having paid work and being less severely injured generally goes along with higher functional capacity, fewer participation restrictions [30], and less stigma in interactions with others, which is likely to increase the possibilities to meet people and be recognized as a potentially attractive partner. Unemployment has been linked to lower marriage rates and higher divorce rates in general population studies [31,32]; the increased psychosocial burden and financial restrictions resulting from unemployment can put a strain on partnerships. The finding that the likelihood of being single increased with increasing time since injury may also reflect a reduction in functional capacity with increasing time since injury. However, this interpretation seems to contradict the observed trend that older participants more often had a partner. Age at injury might play an important role here; people who are older when they sustain SCI may be more likely to be already in a long-term relationship explaining the observed effect of chronological age; on the other hand, living with a partner with SCI may put strain on that relationship leading to an increased likelihood of separation. Longitudinal research is warranted to disentangle the role of age and time since injury more consistently. 

Our study’s findings with regard to living situation are in line with previously reported general population trends [16,33,34]. The result that males more often live alone may partly be due to higher financial independence of males, and traditional gender/family roles of women in many societies, whereby women stay with kin if they are single and only leave the family when they themselves have found a family [33]. The finding that older people and people with less severe injuries more often reported to live alone may be related to the higher self-sufficiency and better access to financial resources that enable independent living in a single household, which is also reflected in the finding that those with higher income more often live alone. In contrast, people with more severe injuries often have an increased need of informal care and therefore a higher likelihood to live with family caregivers. The finding that people in higher HDI countries (mostly Western countries) more often lived alone is congruent to general population trends that indicate marked differences in the prevalence of one-person households between Western countries and countries from other regions of the world [16], although the prevalence of single households is fast growing also in non-Western countries due to labor migration, changing patterns of cohabitation in partnerships, and general lifestyle of the urban middle-class [34]. Next to culturally grown differences, the observed differences according to the HDI level of a country might be explained by the unequal availability of resources needed for independent living. For example, people from higher resource countries might have better access to professional health care and home care services and are therefore less dependent on informal family caregivers and have better access to social security systems supporting them with adapted housing and assistive devices [18]. In many countries with less developed social security systems and restricted access to formal care, living with kin is the only option for people with SCI who need support in their livelihood. 

### Strength and Limitations

This study represents a first cross-country analysis of partnership status and living situation in relation to disability and investigated the relation of a number of individual factors and country-level development with the partnership status and living situation of persons with SCI from different regions of the world. Herein, the study was based on a large sample, providing appropriate power for such analysis. Data were collected with the same standardized questionnaire in all countries and analyzed with state-of-the-art statistical methods. 

Generalizability of findings might however be limited as only eight countries applied random sampling strategies, while 14 countries relied on convenience sampling, thus increasing the risk of sampling bias. It remains also unknown whether cultural, religious, or ethnic issues were related to survey participation in the different countries. Therefore, this study does not provide prevalence estimates for partnership status or living situation of people with SCI, as we cannot claim representativeness of our sample. Our goal was rather to explore predictors of partnership status and living situation of persons with SCI than to provide basic epidemiological data on those outcome variables. Moreover, we solely included basic sociodemographic and SCI characteristics as predictors for the partnership status and living situation at the individual level, and future research is warranted to provide insights into whether health-related issues (e.g., mental health, physical co-morbidities), religion, ethnicity, or country-level characteristics (e.g., social norms related to disability, cultural, or historical peculiarities) impact on the partnership status or living situation of persons with SCI. Further, due to the cross-sectional nature of the data, information on partnership status and living situation represents a snap-shot of people’s situation at the time of the survey, and a life-course approach taking into account the dynamic nature of partnerships and living situations is needed to provide in-depth insights into the lives of persons with SCI. As we do not have information on the duration of the partnership, partnership status before SCI, quality of the partnership, or the satisfaction with the living situation, we cannot assess the health relevance of partnership or living situation. Depending on their quality, partnerships can either provide resources (e.g., social support, reduction in loneliness) or be detrimental for health (e.g., psychosocial strain through conflicts, hostility) [35,36,37]. Moreover, this research does not allow any value judgement on whether having a partner or living with others is positive or negative. It would therefore be important to evaluate whether the partnership status or the living situation is self-determined or whether it is the result of barriers hindering the fulfillment of individual aspirations. A better understanding of people’s aspirations is, however, needed to draw conclusions for policy interventions or rehabilitative services. 

## 5. Conclusions 

This study provides evidence that socioeconomic status and impairment severity at the individual level and country-level development influence the likelihood to live in a partnership and to live with others. Assuming that people’s preferences are otherwise equal on average, these findings likely reflect unequal opportunities. Future research is warranted to shed light on respective preferences and aspirations of persons with SCI and other chronic health conditions. 

## Figures and Tables

**Table 1 ijerph-17-07002-t001:** Basic characteristics of the 12,591 International Spinal Cord Injury (InSCI) community survey participants.

Categorical Variables (% Missing Values in Total Population)	Total
	*N* (%)
**Partnership status** (0.7)	
Single	5119 (41.0)
Having a partner	7380 (59.0)
**Living situation** (1.6)	
Living alone	2351 (19.0)
Living with others	9630 (77.7)
Living in an institution	413 (3.3)
**Gender** (0.3)	
Female	3393 (27.0)
Male	9165 (73.0)
**Age at time of survey in years** (0.6)	
16–30 years	1323 (10.6)
31–45 years	3193 (25.5)
46–60 years	4325 (34.6)
61 years or older	3672 (29.4)
**Employment status** (0)	
Paid work	3880 (30.8)
No paid work	8711 (69.2)
**Injury severity** (4.3)	
Incomplete paraplegia	4155 (34.5)
Complete paraplegia	3381 (28.1)
Incomplete tetraplegia	3284 (27.3)
Complete tetraplegia	1225 (10.2)
**Etiology** (1.6)	
Traumatic	9990 (80.6)
Non-traumatic	2399 (19.4)
**Time since injury in years** (2.7)	
0–5 years	4209 (34.4)
6–15 years	4153 (33.9)
16–25 years	2006 (16.4)
26 years or more	1880 (15.4)
**Human development index ^a^** (0)	
Lowest quartile	2661 (21.1)
2nd lowest quartile	2109 (16.8)
2nd highest quartile	2225 (17.7)
Highest quartile	5596 (44.4)
**Continuous variables**	**Mean (SD); median (IQR)**
Age in years (0.6)	51.3 (15.3); 52 (40–63)
Time since injury in years (2.7)	13.1 (11.9); 9 (4–19)
Education in years (6.1)	12.2 (5.2); 12 (9–15)

Abbreviations: IQR: Inter-quartile range; SCI: Spinal cord injury; SD: Standard deviation. ^a^ Countries in the lowest quartile: Brazil, China, Indonesia, Morocco, South Africa and Thailand; 2nd lowest quartile: Greece, Italy, Lithuania, Malaysia, Poland, Romania; 2nd highest quartile: France, Japan, South Korea, Spain, United States; highest quartile: Australia, Germany, Netherlands, Norway, Switzerland.

**Table 2 ijerph-17-07002-t002:** Associations of individual and country-level characteristics with partnership status: results from unadjusted and adjusted multilevel logistic regressions showing odds ratios (OR) and 95% confidence intervals (95% CI) for being single.

Individual and Country-level Characteristics	Being Single	
	Model 1	Model 2
	OR (95% CI)	OR (95% CI)
***Sociodemographic characteristics***		
** Gender**		
Male	Reference	Reference
Female	1.21 (1.11–1.31)	1.16 (1.05–1.28)
*p*-value	<0.001 ***	0.005 **
** Age (in years)**	0.96 (0.96–0.96)	0.95 (0.94–0.95)
*p*-value	<0.001 ***	<0.001 ***
** Education (in years)**		
Lowest quartile	Reference	Reference
2nd lowest quartile	1.05 (0.95–1.17)	0.99 (0.87–1.12)
2nd highest quartile	1.04 (0.93–1.16)	0.95 (0.83–1.08)
Highest quartile	1.18 (1.06–1.31)	1.12 (0.98–1.28)
*p*-value	0.025 *	0.064
** Net-equivalent household income**		
Lowest quartile	Reference	Reference
2nd lowest quartile	0.57 (0.51–0.64)	0.61 (0.54–0.69)
2nd highest quartile	0.52 (0.47–0.64)	0.55 (0.48–0.62)
Highest quartile	0.48 (0.43–0.53)	0.53 (0.47–0.61)
*p*-value	<0.001 ***	<0.001 ***
**Employment status**		
No paid work	Reference	Reference
Paid work	0.90 (0.83–0.98)	0.68 (0.61–0.76)
*p*-value	0.010 *	<0.001 ***
***Characteristics of the spinal cord injury***		
**Injury severity**		
Incomplete paraplegia	Reference	Reference
Complete paraplegia	1.20 (1.09–1.33)	1.06 (0.95–1.19)
Incomplete tetraplegia	1.06 (0.96–1.17)	1.15 (1.02–1.29)
Complete tetraplegia	1.87 (1.63–2.14)	1.51 (1.29–1.78)
*p*-value	<0.001 ***	<0.001 ***
** Etiology**		
Non-traumatic	Reference	Reference
Traumatic	1.27 (1.15–1.4)	0.89 (0.78–1.01)
*p*-value	<0.001 ***	0.066
**Time since injury (in years)**	1.00 (1.00–1.01)	1.02 (1.02–1.03)
*p*-value	0.008 **	<0.001 ***
***Country-level characteristics***		
**Human Development Index** (1–21, 1 = highest, 21 = lowest HDI)	1.02 (0.98–1.06)	0.99 (0.95–1.03)
*p*-value	0.204	0.598

Model 1: Unadjusted. Model 2: Predictor variables entered simultaneously into the model. Final *N* for model 2 = 10,201. Analysis based on full cases. * *p* < 0.05; ** *p* < 0.01; *** *p* < 0.001; *p*-values from likelihood-ratio tests.

**Table 3 ijerph-17-07002-t003:** Associations of individual and country-level characteristics with the living situation: results from unadjusted and adjusted multilevel logistic regressions showing odds ratios (OR) and 95% confidence intervals (95% CI) for living alone.

Individual and Country-level Characteristics	Living Alone	
	Model 1	Model 2
	OR (95% CI)	OR (95% CI)
***Sociodemographic characteristics***		
**Gender**		
Male	Reference	Reference
Female	1.00 (0.86–1.15)	0.81 (0.69–0.96)
*p*-value	0.975	0.020 *
**Age (in years)**	1.04 (1.04–1.05)	1.04 (1.04–1.05)
*p*-value	<0.001 ***	<0.001 ***
**Education (in years)**		
Lowest quartile	Reference	Reference
2nd lowest quartile	0.91 (0.76–1.09)	0.96 (0.77–1.18)
2nd highest quartile	0.76 (0.62–0.92)	0.83 (0.66–1.05)
Highest quartile	0.98 (0.81–1.19)	1.03 (0.82–1.30)
*p*-value	0.026 *	0.245
**Net-equivalent household income**		
Lowest quartile	Reference	Reference
2nd lowest quartile	0.74 (0.61–0.89)	0.76 (0.61–0.94)
2nd highest quartile	1.29 (1.07–1.56)	1.36 (1.10–1.68)
Highest quartile	1.91 (1.57–2.32)	2.11 (1.68–2.65)
*p*-value	<0.001 ***	<0.001 ***
**Employment status**		
No paid work	Reference	Reference
Paid work	0.92 (0.80–1.06)	0.92 (0.77–1.11)
*p*-value	0.268	0.392
***Characteristics of the spinal cord injury***		
**Injury severity**		
Incomplete paraplegia	Reference	Reference
Complete paraplegia	1.12 (0.95–1.33)	1.19 (0.97–1.46)
Incomplete tetraplegia	0.79 (0.67–0.94)	0.79 (0.64–0.98)
Complete tetraplegia	0.63 (0.51–0.79)	0.75 (0.58–0.98)
*p*-value	<0.001 ***	<0.001 ***
**Etiology**		
Non-traumatic	Reference	Reference
Traumatic	0.78 (0.65–0.94)	0.93 (0.73–1.17)
*p*-value	0.008 **	0.513
**Time since injury (in years)**	1.02 (1.01–1.03)	1.00 (1.00–1.01)
*p*-value	<0.001 ***	0.285
***Country-level characteristics***		
**Human Development Index** (1–21, 1 = highest, 21 = lowest HDI)	0.84 (0.81–0.87)	0.85 (1.00–1.01)
*p*-value	<0.001 ***	<0.001 ***

Model 1: Unadjusted. Model 2: Predictor variables entered simultaneously into the model. Final N for model 2 = 3836 (97% of people having a partner live not alone, 7380 people in partnerships were excluded). * *p* < 0.05; ** *p* < 0.01; *** *p* < 0.001; *p*-values from likelihood ratio tests.

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
