# Peer review of "Partnership Status and Living Situation in Persons Experiencing Physical Disability in 22 Countries: Are There Patterns According to Individual and Country-Level Characteristics?"

_ijerph, 2020, doi:10.3390/ijerph17197002_

Round 1

Reviewer 1 Report

Thank you for this great work. I appreciate the approach using the social model of disability to describe outcomes rather than the disability itself.

Background is well written. One typo, line 84 on page 2, "It thus provides the unique....on partnership and living situations on in..."

Methods:

I think it would make it easier for the reader if you first reported a description of your two data sources in their own section. You could have a header for the overall design then a header for the InSCI data with description of the data set and then a header for the HDI with a description of the data set. Then you could describe your outcome variables and predictor variables from each data source.

Also clarity on how the countries to include in your study would be helpful. I assumed they were chosen as they were in both the InSCI and HDI data sets. You indicate that InSCI was done in 22 countries, but then you only included 21 countries in your analysis. What happened to the 22nd country? Was that country not in the HDI dataset?

Author Response

Thank you for this great work. I appreciate the approach using the social model of disability to describe outcomes rather than the disability itself.

- We thank the reviewer for this positive feedback!

Background is well written. One typo, line 84 on page 2, "It thus provides the unique....on partnership and living situations on in..."

- Thank you for detecting this typo, which we have corrected.

Methods:

I think it would make it easier for the reader if you first reported a description of your two data sources in their own section. You could have a header for the overall design then a header for the InSCI data with description of the data set and then a header for the HDI with a description of the data set. Then you could describe your outcome variables and predictor variables from each data source.

- This is a good suggestion. We have now included a header 'Country-level characteristics' to clarify that the HDI data were macro-level data and were not part of the InSCI data collection (please see p. 4).

Also clarity on how the countries to include in your study would be helpful. I assumed they were chosen as they were in both the InSCI and HDI data sets. You indicate that InSCI was done in 22 countries, but then you only included 21 countries in your analysis. What happened to the 22nd country? Was that country not in the HDI dataset?

- We have now added a paragraph in the methods section on how this international collaboration has been established to clarify how the 22 countries were included (please see p. 3, in yellow):

 "The networks of the International Society of Physical Rehabilitation and Medicine and the International Spinal Cord Society (ISCoS) were used to establish collaboration between countries. National representatives closely involved in SCI research or clinical practice firstly evaluated the feasibility of implementing a survey in their country and then formally joined the InSCI network at the kick-off meeting in February 2015 in Switzerland to start collaboration."

 - Thank you for this feedback on the 21 countries, we realized that this is not clearly described in the current version. We thus added information in the paragraph on HDI on page 4. It is true that the InSCI survey was done in 22 countries and the fact that the HDI-ranking is ranging from 1-21 is due to identical scoring of two countries which were placed on the same rank. We have now included an explanation on this in order to clarify:

"Of note, the HDI score ranges from 1-21 and not from 1-22 as one could assume based on the number of InSCI countries, as Indonesia and South Africa have identical HDI scores and were thus placed on the same rank."

Reviewer 2 Report

This is one of the very few studies providing evidence gathered from several countries to provide an understanding on partnership status and the living situation of persons with a physical disability, therefore I commend the authors for this important contribution. Below I provide some comments that I hope can help add clarity to the manuscript.

Comments

The authors indicate that 12,591 participants were included in the study, I find this sample rather small for 22 countries. The researchers should explain sufficiently why this number, by adding more information on sampling procedure in the methods section.

Line95, “..At least 18 year old community-dwelling people with traumatic or non-95 traumatic SCI”

The age of the oldest participant is not stated. I think it is beneficial to include both the lowest and highest age, even if it probably the authors did not set an age limit before hand.

Line 100 “Sampling frames were defined according 100 to local conditions, including convenience or random sampling.

I think this is a very crucial detail and it should be as elaborate as possible. If convenience random sampling was the only local condition this should be stated, if other factors were considered, they should be added to this section.

Authors should briefly indicate why they used this exclusion criteria, for add clarity.

In order to create comparable categories taking into account country 126 differences in educational and economic systems, we built country-based distributional 127 quartiles for education and income to group persons within countries into four categories 128 ranging from low to high education or income.

How did the authors reconcile these country-based distribution to the international in order to allow for comparison. It would be useful to indicate how different categories varying at a national level were reconciled for all the 22 countries

Line 141 HDI takes into account the life expectancy at birth,

Here the authors should add the word “national” - life expectancy at birth

Line 95 - People with congenital etiology, neurodegenerative disorders (e.g., multiple sclerosis) or Guillain Barré syndrome were excluded.

The authors simply mention these categories were excluded without explaining why. This information should be added.

Author Response

REVIEWER 2

This is one of the very few studies providing evidence gathered from several countries to provide an understanding on partnership status and the living situation of persons with a physical disability, therefore I commend the authors for this important contribution. Below I provide some comments that I hope can help add clarity to the manuscript.

- Thank you very much for the positive feedback! 

Comments

The authors indicate that 12,591 participants were included in the study, I find this sample rather small for 22 countries. The researchers should explain sufficiently why this number, by adding more information on sampling procedure in the methods section.

 - Thank you for the feedback with which we fully agree. We have now added information on the minimal sample size requested by coutries and why it was difficult for some countries to recruit larger samples (please see page 3, lines 113ff.):

"As indicated by a power analysis, countries were requested to recruit a minimum of 200 participants [23]. A total of 10 countries recruited between 200 and 300 participants as detecting potential participants was challenging because SCI is a rare condition and systematic records were missing or relevant databases were not accessible. Recruitment outcomes of the different countries are reported elsewhere in more detail [25]."

Line 95, “..At least 18 year old community-dwelling people with traumatic or non- traumatic SCI” The age of the oldest participant is not stated. I think it is beneficial to include both the lowest and highest age, even if it probably the authors did not set an age limit before hand.

- Thank you, we have now included information on the minimum and maximum of age in our sample (minimum age 18; maximum age 96) in the first paragraph of the results section.

Line 100 “Sampling frames were defined according to local conditions, including convenience or random sampling. I think this is a very crucial detail and it should be as elaborate as possible. If convenience random sampling was the only local condition this should be stated, if other factors were considered, they should be added to this section. Authors should briefly indicate why they used this exclusion criteria, for add clarity.

- We fully agree with the reviewer that the description of the sampling strategies applied in the different countries should be expanded. We therefore added some more explanation on why 14 countries relied on convenience sampling and also added the reference to a paper where this issue has been described in more detail (please see page 3, lines 109ff.):

"Random sampling based on predefined sampling frames was only possible in 8 countries that had access to hospital or patient organization databases. Due to a lack of access to such databases, 14 countries used convenience sampling methods and recruited individuals visiting health care facilities or joining patient organization events [25]."

 Reference 25: Fekete, C.; Brach, M.; Ehrmann, C.; Post, M.W.; Stucki, G. Cohort profile of the International Spinal Cord Injury (InSCI) community survey. Arch Phys Med Rehabil 2020, epub ahead of print.

In order to create comparable categories taking into account country differences in educational and economic systems, we built country-based distributional quartiles for education and income to group persons within countries into four categories ranging from low to high education or income. How did the authors reconcile these country-based distribution to the international in order to allow for comparison. It would be useful to indicate how different categories varying at a national level were reconciled for all the 22 countries

- Indeed, it is difficult to compare socioeconomic indicators of individuals across different countries. As a first step, we created quartiles of education and income for each country separately to differentiate four groups with different educational and income level within a country. As a second step, we combined all the national-level quartiles into one single variable, meaning that all persons in the lowest national category were integrated in the lowest category in the international variable, all persons in the highest national category were integrated into the highest category in the international variable and so on. This variable only gives a rough indication on the educational and income level of people in their respective countries and does not allow for comparison between countries. As we did not intend to compare socioeconomic characteristics of people between countries and only used the variable as predictor for partnership status and living situation, we consider this approach as valid.

Line 141 HDI takes into account the life expectancy at birth, Here the authors should add the word “national” - life expectancy at birth.

- Thank you for this input, which we have integrated in the revised version.

Line 95 - People with congenital etiology, neurodegenerative disorders (e.g., multiple sclerosis) or Guillain Barré syndrome were excluded. The authors simply mention these categories were excluded without explaining why. This information should be added.

- Thank you for this thoughtful feedback. The rationale for this decision is that persons with congenital etiologies, neurodegenerative disorder or Guillain Barré syndrome follow quite different rehabilitation paths and disease Progression is also different than in persons with acquired spinal cord injury. Including people with those health condition would lead to an even more heterogeneous sample. We have now added those arguments to the methods section on page 3, lines 105f.

Reviewer 3 Report

Overall, an interesting topic and research endeavor.

A few comments related to the Introduction: From a critical disabilities standpoint, be cautious and consistent with person- vs identity- first language. For example, you use "Persons with disabilities" in multiple occurrences to begin your paper but use "non-disabled persons" in your first reference to those without disabilities (lines 49,50; p.2). This is especially important to consider in your writing as you explicitly acknowledge that disabled people with SCI may experience stigma and negative perspectives from the general public. There is research and advocacy to point towards the use of identity-first language (ex: Dunn & Andrews, 2015, Person-first and identity-first language: Developing psychologists’ cultural competence using disability language). I wouldn't argue that you need to use identity-first language, only that you stick with either consistently to refer to both disabled/PWD and non-disabled/PWoD.

Notes on Discussion and Strength & Limitations: Given that this a large study across multiple countries, I can accept that the participants do not represent a homogeneous collection of people living from the same cultural, racial, religious, and ethnic backgrounds. However, I do think this needs to be stated in your limitations section, especially considering 15 countries relied on convenience sampling (Side note, unless I missed it, were there 22 or 23 countries surveyed? Because on line 300, 301, p. 10 you have 8 countries as randomly sampling and 15 as utilizing convenience sampling for a total of 23). These 15 countries relying on convenience sampling inherently suggests a limitation of cultural and other sociological factors impacting recruitment, research literacy, and/or incentive to participate thus skewing your findings. Cultural considerations such as histories of Eugenics, Neoliberal Rationales, and general differences in disability ideology/social perspectives should be added to your Limitations suggestions and help to further parameter your Discussion section's assumptions/hypothesis/pattern interpretation of the findings. These variables vary across countries as well as within geographical and corners of the countries themselves and would present a deeper/more in-depth defense of partnership/co-habitative predictors in future studies.

Looking forward to following future studies (both qualitative and quantitative endeavors) to further explore and construct pattern formations of partnership and/or co-habitation for disabled people with SCI.

Author Response

Overall, an interesting topic and research endeavor.

- Thank you for this positive feedback!

A few comments related to the Introduction: From a critical disabilities standpoint, be cautious and consistent with person- vs identity- first language. For example, you use "Persons with disabilities" in multiple occurrences to begin your paper but use "non-disabled persons" in your first reference to those without disabilities (lines 49,50; p.2). This is especially important to consider in your writing as you explicitly acknowledge that disabled people with SCI may experience stigma and negative perspectives from the general public. There is research and advocacy to point towards the use of identity-first language (ex: Dunn & Andrews, 2015, Person-first and identity-first language: Developing psychologists’ cultural competence using disability language). I wouldn't argue that you need to use identity-first language, only that you stick with either consistently to refer to both disabled/PWD and non-disabled/PWoD.

- Thank you for raising this important issue. We have now revised the terminology throughout the manuscript and have changed it into persons with disabilities or PwoD.

Notes on Discussion and Strength & Limitations: Given that this a large study across multiple countries, I can accept that the participants do not represent a homogeneous collection of people living from the same cultural, racial, religious, and ethnic backgrounds. However, I do think this needs to be stated in your limitations section, especially considering 15 countries relied on convenience sampling (Side note, unless I missed it, were there 22 or 23 countries surveyed? Because on line 300, 301, p. 10 you have 8 countries as randomly sampling and 15 as utilizing convenience sampling for a total of 23). These 15 countries relying on convenience sampling inherently suggests a limitation of cultural and other sociological factors impacting recruitment, research literacy, and/or incentive to participate thus skewing your findings. Cultural considerations such as histories of Eugenics, Neoliberal Rationales, and general differences in disability ideology/social perspectives should be added to your Limitations suggestions and help to further parameter your Discussion section's assumptions/hypothesis/pattern interpretation of the findings. These variables vary across countries as well as within geographical and corners of the countries themselves and would present a deeper/more in-depth defense of partnership/co-habitative predictors in future studies.

Looking forward to following future studies (both qualitative and quantitative endeavors) to further explore and construct pattern formations of partnership and/or co-habitation for disabled people with SCI.

- We fully agree with the Reviewer that the differences in terms of cultural, ethnical or religious backgrounds should be acknowledged in the limitations section and have therefore included the following statements (please see p. 10, lines 327ff. and 336ff.):

 "It remains also unknown whether cultural, religious or ethnic issues were related to survey participation in the different countries. Therefore, this study does not provide prevalence estimates for partnership status or living situation of people with SCI as we cannot claim representativeness of our sample."

 "Moreover, we solely included basic sociodemographic and SCI characteristics as predictors for the partnership status and living situation at the individual level, and future research is warranted to provide insights into whether health-related issues (e.g., mental health, physical co-morbidities), religion, ethnicity, or country-level characteristics (e.g., social norms related to disability, cultural or historical peculiarities) impact on the partnership status or living situation of persons with SCI."

- Moreover, we thank the reviewer to detect an important mistake, as there were in fact 14 countries recruiting via convenience sampling and not 15. We have corrected this.